# Comparative Analysis of G-Layers in Bast Fiber and Xylem Cell Walls in Flax Using Raman Spectroscopy

**DOI:** 10.3390/biom13030435

**Published:** 2023-02-24

**Authors:** Anne-Sophie Blervacq, Myriam Moreau, Anne Duputié, Simon Hawkins

**Affiliations:** 1Université de Lille, Sciences et Technologies, CNRS, UMR 8576-UGSF-Unité de Glycobiologie Structurale et Fonctionnelle, F-59000 Lille, France; 2Université de Lille, Sciences et Technologies, CNRS, UMR 8516-LASIRE-Laboratoire de Spectroscopie Pour les Interactions, la Réactivité et l’Environnement, F-59000 Lille, France; 3Université de Lille, Sciences et Technologies, CNRS, UMR 8198-EEP-Evo-Eco-Paléo, Bâtiment SN2, F-59000 Lille, France

**Keywords:** bast fibers, Raman chemical imaging, flax, G-layer, gravitropic stress, confocal Raman spectroscopy, tension wood, opposite wood

## Abstract

In a response to gravitropic stress, G-layers (gelatinous layers) were deposited in xylem cell walls of tilted flax plants. G-layers were produced in both tension wood (upper side) as expected but were also observed in opposite wood (lower side). Raman spectral profiles were acquired for xylem G-layers from the tension and opposite side as well as from the G-layer of bast fibers grown under non-tilted conditions. Statistical analysis by principal component analysis (PCA) and partial least square-discriminant analysis (PLS-DA) clearly distinguished bast fiber G-layers from xylem G-layers. Discriminating bands were observed for cellulose (380–1150–1376 cm^–1^), hemicelluloses (517–1094–1126–1452 cm^–1^) and aromatics (1270–1599–1658 cm^–1^). PCA did not allow separation of G-layers from tension/opposite-wood sides. In contrast, the two types of xylem G-layers could be incompletely discriminated through PLS-DA. Overall, the results suggested that while the architecture (polymer spatial distribution) of bast fibers G-layers and xylem G-layers are similar, they should be considered as belonging to a different cell wall layer category based upon ontogenetical and chemical composition parameters.

## 1. Introduction

The cell wall is a characteristic structural feature of plants that undergoes dynamic remodeling during developmental differentiation. Flax (*Linum usitatissimum*) is an interesting plant species for investigating cell wall biology, as the stem contains different cell types with either cellulose-rich or lignified-rich cell walls [1,2,3]. Cellulose-rich walls are found in stem outer-tissue bast fibers (BFs) while lignified-enriched walls are found in the xylem tissue located in the inner-stem.

The flax BFs originate from the procambium and are believed to function as plant “muscles” [3,4] that contribute to the mechanical support in the slender stem of this species. Similar procambial fibers (phloem fibers) are also observed in many other fiber species, including hemp (*Cannabis sativa*), nettle (*Urtica dioica*) and ramie (*Boehmeria nivea*) [5].

Different studies have shown that the cell walls of flax BFs contain 70% cellulose with a low microfibril angle, 5–15% non-cellulosic polysaccharides consisting of β 1,4 galactans and arabinogalactan proteins, and in contrast to flax xylem cells, very low amounts of lignin [1,5,6]. Low proportions of xyloglucan hemicelluloses are also present [5,6] and highly substituted xylan epitopes have been identified in the middle lamella/primary cell wall with LM11 antibodies [6]. The CAZy categorization of extracted cell wall proteins in flax clearly shows the importance of hemicellulose metabolism in BF-rich outer stem tissues [6].

From a structural point of view, the flax BF cell wall is composed of the following different layers (from the exterior to the cell lumen): middle lamella, primary cell wall (PCW), secondary cell wall (SCW) and an inner gelatinous layer (G-layer) that is synthesized after SCW formation [3,7]. In contrast, xylem cell walls in flax and other species do not usually contain a G-layer. Nevertheless, it has been known since 1860 that a gelatinous cell wall layer is present in the so called “reaction wood” of Dicots [8,9] formed in response to gravitational stimulus. This gelatinous cell wall layer is produced de novo in the xylem and is deposited inside the lignified secondary cell wall. It is known to be highly cellulosic [10,11] and is called the G-layer for gelatinous, due to its gel-like appearance. 

It has also been demonstrated that G-layers can be sometimes found in xylem cells of inclined flax plants [12]. In this study, a comparison of the composition of G-layers from flax BFs and xylem cells using antibodies targeted against xylan/arabinoxylan (LM11) and galactan (LM5) cell wall epitopes showed that both structures contained galactans. However, as shown by a previous study [6] flax BF cell walls also react to anti-xyloglucan antibodies (LM15, LM24) as well as anti-mannan/heteromannan antibodies (LM21) and it would obviously be of interest to obtain information about the presence/absence of different cell wall polymers in flax G-layers. Nevertheless, because of the structural similarity of the flax BF G-layer and the G-layer present in tension wood xylem cells, it has been proposed that these cell layers should be considered as belonging to a new cell wall layer category called the “tertiary cell wall” [3,11,13,14,15,16]. This concept is however still debated since the G-layer of the flax BF cell wall is deposited while the cell is still living, and other workers therefore consider the BF G-layer to be a specialized region of the secondary cell wall [16].

In order to see how closely the composition and structure of these two cell wall layers (BF G-layer and xylem G-layer) resemble each other, we decided to use Raman spectroscopy. This approach has been widely used to characterize cell wall structure in plants [17] and offers a number of advantages over immunochemical-based imaging since the information obtained about cell wall polymers is not restricted to the antibodies available and/or used in the particular study considered. We have previously used Raman spectroscopy to investigate changes to cell wall composition and polymer distribution in bast fibers of flax plants exposed to a gravitational stress [18]. In this paper we report the use of a similar approach to investigate BF and xylem G-layers in flax, thereby providing more complete information about the comparative composition of G-layers from flax BFs and xylem cells and contributing to the on-going tertiary cell wall debate.

## 2. Materials and Methods

### 2.1. Plant Material and Gravitropic Stress Induction

Seeds of fiber flax (*Linum usitatissimum* L., var. Diane) were germinated in a greenhouse under natural light conditions with daily watering. Plants were exposed to gravitropic stress, as previously described in [18]. Briefly, 6-week-old plants were maintained tilted at 45° with bamboo tutors for the last six weeks of their growth (total: 12 weeks). 

### 2.2. Sample Fixation, Sectioning and Histochemistry

Stem samples (1 cm long) were collected from the middle region of the stem (10 cm above the cotyledon scar) from ten individual control and ten individual stressed plants. Samples were dehydrated and embedded in Technovit^®^ 7100 (HEMA: 2 hydroxyethyl methacrylate, Kulzer). Five μm thick cross sections were obtained with a Leica microtome RM2065.

Histochemistry was performed with toluidine blue O (global observation) in a Wiesner reaction (phloroglucinol/HCl). Aromatics and lignin were localized through photonic optical epifluorescence [λ_exc_ = 405 nm and λ_em_ = 425–475 nm]. Calcofluor white was used to confirm the cellulosic nature of cell walls [λ_exc_ = 405 nm and λ_em_ = 425–475 nm].

For Raman microspectroscopy, samples were embedded in PEG (polyethylene glycol) 1500, as previously described [17]. Transversal sections (30 μm) were obtained with a Leica microtome. PEG was removed by water baths and sections were air-dried prior to mounting in bidistilled water. Standard glass slides and glass coverslips were used.

### 2.3. Immunolabeling

Technovit 7100 was removed in acetone, then sections were rehydraded in a 5 min graded ethanol series, and finally in phosphate buffer saline (PBS: 137 mM NaCl, 2.7 mM KCl, 20 mM Na_2_HPO_4_, 1.5 mM KH_2_PO_4_). Non-specific sites were saturated in 2 × 15 min PBST (2% BSA (*w*/*v*) and 0.05% tween 20 (*v*/*v*) in PBS) followed by rabbit primary antibodies incubations in PBST (dilutions from 1/10), for 1 h 30 at 37 °C. Immunolocalisation of arabinan (α-1, 5-L-arabinan) and xylan epitopes were achieved on 5 µm semi-thin sections using rat monoclonal antibodies. LM6 is an anti-arabinan monoclonal antibody (antigen: pectic polysaccharide/α 1, 5 arabinan, epitope: linear α 1, 5 arabinan, provided Megazyme, Bray, Ireland). LM10 is an anti-xylan monoclonal antibody (antigen: heteroxylan, epitope: unsubstituted β-1, 4 xylan, provided by Megazyme, Bray, Ireland). They were both diluted 1/10 in PBST buffer (137 mM NaCl, 2.7 mM KCl, 16.3 mM Na_2_HPO_4_, 1.5 mM KH_2_PO_4_, pH 6.8, 2% (*w*/*v*) BSA, 0.01% (*v*/*v*)Triton X100). Secondary incubation was carried out with goat anti-rat antibody, conjugated with fluorescein isothiocyanate (FITC, Jakson immunoresearch) diluted 1/100 in PBST. Observations were performed on a photonic epifluorescent Olympus BH-2 microscope [λ_exc_ = 495–499 nm and λ_em_ = 515–521 nm].

### 2.4. Morphometric Data Acquisition and Treatment 

Xylem G-layer thickness was measured for the tension side and the opposite side, excluding the xylem SCW. A total of 387 measures (211 and 176 respectively for tension wood and opposite wood) were acquired from the same 3 cross sections/stem. Acquisitions were carried out with FIJI software (https://imagej.net/software/fiji/ accessed on 20 December 2022).

A bilateral t-test was used to test for differences in G-layer thickness after having checked for homoscedasticity using Bartlett’s test.

### 2.5. Raman Microspectroscopy: Spectral Acquisition

Analyses were performed on a LabRam HR-Evolution (Horiba scientific), as previously described [18]. Three stems (=individual) were used. For each stem, 3 cross sections were taken. A minimum of four spectra were acquired for each category (BF G-layer, xylem opposite G-layer, xylem tension G-layer). For one category, we pooled 4 spectra × 3 cross sections × 3 individuals, i.e., a minimum of 36 spectra.

The total number of spectra acquired was respectively *n* = 36 spectra for xylem G-layer from the opposite side, *n* = 40 for xylem G-layer from the tension side and, finally, *n* = 36 spectra for bast fiber G-layer from flax cultivated in optimal culture conditions.

For a simple comparison with a lignified cell wall, 20 spectra were acquired in one cross section through young wood (control plant) within the first 5 cell layers from the vascular cambium. Fiber-tracheids and large vessels, anticlinal and periclinal cell walls were used to acquire spectra.

Average spectra were then normalized on the 380 cm^–1^ peak assigned to cellulose. This peak was chosen according to previous work on flax BFs [18]. Briefly, excitation was performed at 515 nm, with an entrance slit of 100 µm, and equipped with a 600 lines/mm^−1^ grating. The laser spot was 0.8 µm. Spectra were acquired after 5 min of bleaching, 100 accumulation/sec, from 300 cm^–1^ to 1800 cm^–1^, corresponding to the plant cell wall fingerprint region.

### 2.6. Unsupervised Exploratory Analysis: Principal Component Analysis and Hierarchical Clustering of Spectra

Principal component analysis (PCA) was used to identify differences among spectra and to locate peaks contributing most to among-sample variation. PCA is a non-supervised method that helps visualize the structure of the variability in highly dimensioned data. Spectra were pre-processed prior to PCA as follows: (i) peaks < 300 cm^–1^ and >1750 cm^–1^ were suppressed for all spectra, (ii) the baseline was corrected using polynomial fitting (degree 6), (iii) spectra were normalized by equalizing the area under each spectrum. 

To determine whether spectra differed between the three categories, they were classified using hierarchical cluster analysis (HCA), a non-supervised method that recursively partitions a dataset according to dissimilarities. Here, Euclidean distances were used among pre-processed pairs of spectra.

### 2.7. Supervised Classification of the Three Categories of Spectra

Partial least square-discriminant analysis (PLS-DA) was used to determine whether the Raman spectra of the three categories of the cell wall layer could be identified. Due to the large number of variables, a PLS-DA using all data was always able to discriminate among the three categories of samples because of overfitting.

Therefore, to test whether the three types of spectra could be identified, PLS-DA was performed with double cross-model validation, with a 7-fold cross validation for the external loop and a 6-fold cross-validation for the inner loop. This procedure was repeated 100 times. Spectra were therefore predicted to belong to BF G-layers, opposite side xylem G-layers or tension side xylem G-layers, sometimes with errors. All 100 confusion matrices were collected and averaged to assess the risk of misclassification (and thus the degree of identifiability of the three categories). 

Finally, a permutation test was applied to assess the significance of the differences.

To better investigate the differences among BF G-layers and G-layers of both poles, these analyses (PCA, HCA and PLS-DA) were repeated on three datasets: (i) the full dataset; (ii) a dataset excluding aromatic wavenumbers (1270–1421 cm^–1^, and above 1510 cm^–1^), because these wavenumbers were found to differ between BF G-layers and xylem G-layers; (iii) a dataset excluding bast fiber G-layers, to focus on differences between xylem G-layers of both sides.

PCA, HCA and PLS-DA were performed with R software (https://www.r-project.org/ using libraries ChemoSpec and mixOmics accessed on 20 December 2022) [14,19].

### 2.8. Raman Chemical Imaging

Raman chemical imaging was performed using major individual peak wavenumbers identified as differing among cell wall types. To enable comparison, a ratio was used based on the intensity of the reference cellulose peak at 380 cm^–1^ (area under the peak) for each pixel of the image. 

## 3. Results

### 3.1. Lodging Induces G-Layer in Flax Wood

As expected, tilted flax stems contained xylem tissue with G-layers, both in young (close to the vascular cambium) and mature wood (close to the pith) in both the tension side (Figure 1E) and opposite side (Figure 1G). The first characterization was performed using classical histochemistry stains and immunolocalization (Figure 1D–M). 

Flax xylem exhibits three cell types: ray cells (parenchyma), small diameter fiber-tracheids and large vessels [20]. Xylem G-layers appeared in aligned fiber-tracheids (groups of 4–5 adjacent cells) and rarely in single ray cells (Figure 1E–G). No G-layer was observed in large vessels. In wood from tilted trees, G-layer appears only in the tension side [21], both in differentiating vessels and fibers [21]. However, in flax submitted to this long-term gravitropic stress, we also detected such a G-layer in young and mature wood from the opposite side (Figure 1G). In contrast, G-layers were only noticed very rarely in control plants (data not shown). 

In flax, xylem G-layers are mainly cellulosic in nature as no lignin was observed after the Wiesner reaction (phloroglucinol/HCl, Figure 1B,D,F,H). In the same manner, arabinan and xylan epitopes were also localized with immunolocalization in xylem G-layers (Figure 1L–M). Contrary to observations in tree tension wood, xylan epitopes were localized as diffuse spots or rafts, but only in xylem SCW (Figure 1M). Some weak labeling could also be seen in the innermost part of the G-layer. As in tree tension wood, we also localized arabinan epitopes in the innermost layer of the xylem G-layer (Figure 1L) and as rafts within ray cells. 

The average thickness of the flax xylem G-layer was 3.70 ± 1.1 μm in the tension side and 3.69 ± 0.89 μm in the opposite side, a non-significant difference (*t* = 0.125, df = 375; *p*-value = 0.90; *n*= 387 measures covering all categories). According to this morphometric criterion, xylem G-layer thicknesses seem very similar regardless of whether they are formed in tension wood or opposite wood.

### 3.2. G-Layer and Bast Fibers Spectra 

Raman spectra were acquired specifically on the G-layers, but not SCWs, of xylem cells in tension wood and opposite wood, as well as on the G-layers of control bast fibers (BFs) or control wood SCWs (Figure 2). For control wood (from plants grown under optimal conditions of culture), spectra were acquired in young xylem close to the cambial zone (fiber-tracheids, large vessels, both for periclinal/anticlinal cell walls).

The average spectra of the G-layers seemed very similar regardless of their origin (Figure 2A). Nevertheless, subtle differences could be observed between xylem G-layers and BF G-layers (Appendix A). Aromatics detected within the G layer (a1–a4) could not be directly linked to lignin as this layer gave a negative reaction with the Wiesner reagent (Figure 1F–H). In contrast, the control wood average spectra shows a different spectral profile (Figure 2B, notably considering amorphous cellulose peaks, or lignin peaks intensities). In wood, the aromatic compound peaks could be related to lignin polymer according to the Wiesner reaction (Figure 1B,D,F,H) and autofluorescence (Figure 1I).

These spectral profiles confirmed the cellulosic nature of xylem G-layers. Peaks of xylan and xyloglucan are clearly present both in xylem and BF G-layers. 

Despite the high degree of similarity between the xylem and BF G-layer spectra, some differences were observed. The xylem, but not BF, spectra showed peaks assigned to the lignin monomer G-unit (a1 peak, 1270 cm^–1^) and aromatics (a2 peak, 1421 cm^–1^). Spectral assignations also suggested that xylem G-layers contained higher amounts of coniferyl aldehydes and alcohols (a3-a4 peaks, 1599–1658 cm^–1^) compared to BFs. Bands attributed to cellulose (607–900 cm^–1^) and hemicelluloses (1729 cm^–1^) also showed higher intensities than corresponding bands in BFs.

In order to further evaluate potential similarities/differences between xylem and BF G-layers, spectral data were analyzed by PCA (principal component analysis) (Figure 3, Appendix A). Given that no lignin was detected with the Wiesner Reaction in all G-layers, the term “aromatics” was used for the bands related to [a1: 1270 cm^–1^; a2: 1421 cm^–1^; a3-a4: 1599–1658 cm^–1^].

The PCA biplot (Figure 3A) clearly showed that the PC1 (explaining 54% variability) axis separates the datasets into two clusters: xylem G-layers (red and green dots) and BF G-layers (blue dots). The left cluster contains xylem G-layers from the tension side (red dots) and the opposite one (green dots). No combination of other PCs clearly separated the spectral features from both xylem G-layers. The examination of PC1 loadings (Figure 3B) highlighted significant differences between BF and xylem G-layers, with positive contributions (characteristic of BFs) by bands corresponding to cellulose (380–405–435–562–900–1094–1126–1150 cm^–1^), β-glucan (969 cm^–1^), xylan and XG (490–517–1094–1126 cm^–1^). Negative contributions (characteristic of xylem G-layer) corresponded to bands assigned to cellulose (1376 cm^–1^), hemicelluloses ( 1452 cm^–1^) and aromatics (1270–1421 and 1599–1658 cm^–1^).

The comparison of average spectra (Figure 2 and Appendix A, Appendix A) also indicated the absence and/or reduced intensity of bands corresponding to aromatic compounds in BF G-layers. In an attempt to evaluate whether these compounds were the only cause of the observed differences, the corresponding bands (1270–1421, and all peaks >1510 cm^–1^) were removed from the data set and the PCA repeated (Figure 3C,D). As observed previously (Figure 3A), there was still a clear separation along the PC1 axis between BF and xylem G-layers, and no other PCs allowed for discriminating the xylem G-layer from opposite (green dots) and tension (red dots) sides. This partial analysis demonstrated that the presence/absence of aromatics was not the only reason for differences between BF and G-layer spectra. The examination of the corresponding discriminant zones (Figure 3D) confirmed the involvement of cellulose (380–405–435–1000–1126–1150–1376 cm^–1^) and xylan/XG/cellulose (490–517–1094–1126 cm^–1^) in distinguishing BF and xylem G-layers.

Finally, a PCA was achieved only on xylem G-layer spectral data in order to compare xylem G-layers from tension wood and opposite wood cells. Our results (Figure 3E,F). suggested a possible separation along PC2 (explaining 8.1% variability) even though most of the data were nevertheless mixed. Only very few peaks distinguished these two types of G-layers (Figure 3F). These peaks were assigned to cellulose (380–1150 cm^–1^), xylan/XG/cellulose (1091–1126 cm^–1^) and aromatics (1599–1658 cm^–1^).

Such differences between xylem and BF G-layers were also confirmed by focusing on specific spectral zones [360–440 cm^–1^] for cellulose, [500–700 cm^–1^] for xylan, [900–1230 cm^–1^] for xylan/XG and cellulose, and [1240–1430 cm^–1^]-[1500–1730 cm^–1^] for aromatics (Appendix A).

The results were confirmed by a hierarchical clustering analysis of samples. BF samples were clearly separated from xylem samples (Figure 4D). When the same analysis was performed using only spectra from the xylem G-layers, the G-layers from opposite and tension sides were not separated by the clustering. 

### 3.3. PLS-DA of Spectral Datasets

Whether using the full dataset (Figure 4A) or modified datasets (without aromatic bands; Figure 4B), PLS-DA biplots (component 1 vs. component 2) clearly discriminated BF G-layers (blue) from xylem G-layers (green, red), notably along the PLS-DA 1 axis. The fact that BFs were still separated from xylem G-layers in the modified dataset (Figure 4B) indicates that aromatics are not the only factor contributing to such a separation. The second axis PLS-DA 2 partially separated xylem G-layers from both wood sides, but overlapping was nevertheless observed. 

When trying to discriminate only xylem G-layers spectra (Figure 4C), the PLS-DA was less successful. There were some misclassifications, and regardless of the dataset used, these misclassifications concerned almost only tension versus opposite xylem G-layers (Table 1). This indicates that the composition of these two types of G-layers are barely distinguished (while being clearly distinguishable from the BF G-layer).

In conclusion, flax xylem G-layers possess specific spectral profiles that are statistically different from those of BFs. PCA and PLS-DA established that the presence of aromatics in G-layer spectra did not explain all of the observed differences with BF spectra. Although G-layer spectra from tension wood and opposite wood poles share very similar spectral profiles, PLS-DA showed that major particularities exist between these two poles. Overall, the combined use of PCA and PLS-DA proved to be a rapid and reliable approach, demonstrating that BF and xylem G-layers can be separated using Raman spectroscopy.

### 3.4. Xylem G-Layer Chemical Imaging

Raman chemical imaging of peaks showing major contributions (positive or negative) to PC1 was undertaken to provide more detailed information about the distribution of cell wall polymers in flax xylem G-layers (Figure 5). The results confirmed the cellulosic nature of the xylem G-layer, but also indicated an irregular distribution of significant chemical bonds within the G-layer (Figure 5). Deformed fiber-tracheids containing a deformed G-layer are richer in these chemical bonds (in Tension side).

Amorphous cellulose (most of the cellulose peaks below 1334 cm^–1^) is particularly present as concentric circles within all G-layers. Crystalline cellulose (1376 cm^–1^) gave a similar characteristic spatial distribution. For xylan/XG, spatial distribution patterns are similar for the G-layer but are different between poles in other surrounding xylem cell types, as observed for cellulose. 

In conclusion, xylem G-layers exhibited a quite similar pattern of distribution, whatever the nature of cell wall polymers considered, even if the opposite side seemed less affected. Cellulose and hemicelluloses are intimately intricated (according to superposed spatial chemical distributions). In the tension side, a depletion of the amounts of hemicellulose (1452 cm^–1^) is noticed in fiber-tracheids and large vessels (notably periclinal cell walls). Moreover, according to the intensities, it was twice as high in the tension side compared to the opposite side.

### 3.5. Induction of De-Lignification in Surrounding Xylem Tissues

According to the literature, lignin is either absent, partially abundant or highly abundant within xylem G-layers depending on the plant species considered. In order to evaluate the presence/absence of lignin/aromatics in flax xylem G-layers, we performed Raman chemical imaging (Figure 6). The peak related to a2 (1421 cm^–1^) was too minor to be imaged (Figure 2, Appendix A).

Raman chemical imaging established that no lignin or aromatics are present in both G-layers in flax as was suggested by histochemistry (Figure 1).

The examination of our results showed an interesting effect of the presence of G-layers in xylem. Lignification is lower in the different xylem cell types surrounding this G-layer zone. Ray cells also exhibited less cellulose and xylan/XG, and fiber-tracheids/large vessels are deprived of lignin. A similar effect was not noticed on the opposite side. 

In conclusion, long-term gravitropic stress induces cell wall remodeling, resulting not only in the production of a G-layer in flax xylem cells, but also in important changes to the structure of walls in immediately neighboring cells. Such changes are different depending upon the pole (tension vs. opposite) considered, suggesting the differential regulation of cell wall metabolism. 

## 4. Discussion

Plants are immobile organisms and have therefore had to develop sophisticated mechanisms for dealing with environmental stress and its consequences. Among them, the phenomenon of lodging in which plant stems are bent parallel to the ground as a result of high wind/rain is observed in many crop species including flax. We have previously use Raman spectroscopy to characterize the effects of tilting on the composition of flax bast fiber cell walls in an experimental system specifically designed to separate gravitropic and autotropic effects [18]. In this paper, we use a similar approach to compare the spatial distribution of cell wall polymers in flax bast fibers with that of the G-layer formed in the tension wood and opposite wood of flax xylem in response to a gravitropic stress. 

In 2012, the term “*tertiary cell wall*” was coined to describe both the cellulose-rich cell wall (G-layer) of flax fibers and the G-layer (gelatinous layer) deposited interior to the secondary wall in the xylem of flax and trees in response to gravitropic/autotropic stress [3,13]. However, from an ontogenetical point of view, the description of fiber/G-layer cell walls as tertiary cell walls is open to debate and other workers consider them to be a particular layer of the secondary cell wall since they are not deposited post-mortem [16]. In our work, we observed that G-layers are formed in both young (i.e., close to the vascular cambium) and older (close to the pith) cells of flax tension wood and opposite wood. The fact that G-layers are formed in young differentiating cells in which secondary cell wall lignification is still occurring, as well as in older cells, in which secondary cell wall formation has been completed, would argue against the use of the term “tertiary” to describe flax G-layers on a strictly ontogenetic basis. 

Both flax BFs and xylem G-layers are involved in the mechanical support of the plant, and it has therefore been proposed that they will show the same architecture and chemical features: cellulose microfibrils aligned with the cell axis and a thick layer/wall devoid of lignin and xylan, but containing Rhamnogalacturonan I. In order to test this hypothesis, we used Raman spectroscopy to characterize and compare the chemical composition of tension wood and opposite wood xylem G-layers and bast fiber G-layers in flax. 

Principal component analysis (PCA) and PLS-DA of acquired spectra showed that xylem and BF G-layers can be clearly separated. Specific spectral zones responsible for the separation included cellulose [360–440 cm^–1^], xylan [500–700 cm^–1^], xylan/xyloglucan/cellulose, [900–1230 cm^–1^] and aromatics [1240–1430 cm^–1^]-[1500–1730 cm^–1^]. In contrast, the use of unsupervised methods was not able to clearly separate xylem G-layers from flax tension and opposite woods. PLS-DA resulted in a number of classification errors, and it is possible that the observed incomplete separation could be due to differences in band intensity levels.

Data from the Raman average spectra, PC loadings and Raman chemical imaging all indicated the presence of hemicelluloses (xylan and xyloglucan) in the G-layers of bast fibers and xylem cells (this study and [18]). While such an observation is in contrast to a previous paper proposing that xylan is absent in the tertiary cell wall [11], our spectroscopic data was also supported by using the LM10 anti-xylan antibody. Peaks corresponding to xyloglucan overlapped those of cellulose in spectra (Appendix A). However, the presence of this type of hemicellulose in flax cell walls (xylem and baster fibers) has been confirmed in several previous studies [6]. These results are also in agreement with previous observations on the xylem G-layer from Japanese hardwoods [30] that localized both xylan and lignin. Low amounts of the latter polymer have been identified in the middle lamella, primary cell wall and S1 secondary cell wall layer of flax bast fibers [1,20]. In contrast, no bands specifically assigned to lignin could be identified in our study of flax xylem G-layers. The absence of lignin was previously noticed in isolated poplar G-layers [21,31], whereas lignin, as traces, was noticed in G-layers from the tension woods of different trees [30,32]. Globally, in trees, the xylem G-layer is reported to be unlignified, partially-lignified or fully lignified depending on the species [30].

In flax, differences in the presence of aromatics (1270–1599–1658 cm^–1^) were also observed in xylem G-layer average spectra compared to bast fibers. However, such differences were not obviously present in corresponding Raman chemical imaging. This apparent discrepancy might be due to the surrounding chemical environment resulting in high intensities on spectra. Despite the presence of these aromatics noticed in the average spectra, both PCA and PLS-DA continued to separate bast fibers and xylem G-layers even in modified spectra with removed aromatic bands. Such an observation shows that the presence/absence of lignin and/or aromatics are not the only differences in cell wall composition between these different cell types. 

The examination of the spatial distribution of major interesting peaks in xylem G-layers (Figure 5) showed a progressive (green pixels close to the middle lamella → yellow→ red/orange pixels) and symmetrical distribution (green pixels close to the lumen). Such a progressive and symmetrical abundance (as concentric layers) is probably due to the ontogeny of this G-layer. It is most likely constituted in a non-autonomous manner in these differentiated xylem cells (i.e., by diffusing precursors from the middle lamella, through primary, then the different secondary cell wall layers). Its deposition as a *post mortem* event is compatible with the later deposition of extractives such as terpenoids, alkaloids or phenolic compounds that could explain the observed autofluorescence of the innermost part of the G-layer as well as the aromatic contributions in the spectra [16].

In flax fiber-tracheids (xylem), G-layer is therefore produced during the secondary cell wall patterning, and/or will replace/alternate with the S3 layer [16,33,34]. We can also suggest that such progressive intensity and symmetrical depositions could help the stressed plant to mechanically recover a vertical position according to its gelatinous status. The study [16] suggested that such a G-layer exhibited high mesoporosity and the ability to generate a large contraction of the layer along the axis fiber. If the microfibril angle is abruptly changed, such a G-layer could detach easily from the S2 layer (i.e., from xylem cell wall) and induce a release of the high tensile stress.

Raman chemical imaging also suggested the existence of cell-specific changes in the lignification process in flax tension wood and opposite wood within the vicinity of G-layers. The presence of a G-layer (particularly in tension wood) reduced the intensities of lignin-related bands in neighboring xylem cells. This observation suggests that the formation of a G-layer induces localized hypolignification within a distance of 10–20 μm all around the aligned fiber-tracheid cells containing the G-layers. In literature, it is known that lignin metabolism is switched off when xylem G-layers are produced in tension wood [16]. This could be the result of transcriptional regulation via MYB transcription factors such as *MYB58* and *MYB 63* [35] and *MYB4* in flax BF [36] and hemp [37]. In compression wood in the conifer *Picea abies*, the lignification pathway was particularly modified due to the regulation of peroxidases and laccases [38]. 

With regards to cell wall architecture, the “gelatinous structure” that characterizes bast fibers and the xylem G-layer in flax is assumed to be a common feature [12,39,40]. Our Raman chemical imaging showed that the spatial distribution of significant chemical bonds was similar and occurred as concentric layers, with higher intensities present in the inner part. The observed co-spatial distribution of significant bands related to cellulose, or xylan/xyloglucan might be related to the entrapment of aggregated Rhamnogalacturonan I with laterally interacting cellulose microfibrils [3]. 

Although Raman hyperspectral profiles and chemical imaging are powerful approaches to investigate cell wall structure, they do have research limits. Imaging is often applied to obtain spatially resolved chemical information about complex samples such as plant stem sections (histological level). Each pixel collected from such an image (i.e., chemical map) corresponds to an independent spectra [41]. Univariate analysis may provide information, but it does not take into account interference effects of spectral overlap, background interference, fluorescence and laser power fluctuations. PCA methodology allows the exhaustive characterization of the hyperspectral datasets [42]. In this study, chemical imaging was performed after the PCA approach in order to restrict chemical imaging only to positive or negative contributions. 

In addition to its fundamental interest, a better understanding of cell wall remodeling in response to gravitropic mechanical stress is also of practical interest. In flax for example, the composition of bast fiber cell walls is greatly altered in the tension pole [18], thereby impacting their quality for textile uses. In conifer wood, stress-induced gaps between cell layers have been reported that could alter tensile properties, and the modification of the global sugar composition of lignified cell walls will affect bioethanol production (see references in [16]). Such alterations could be either negative, as for beech wood dedicated to pulp, carpentry or furniture production in which the presence of tension wood is considered as a major default [43], or conversely, G-layer is expected in willow wood in order to increase the glucose ratio for biofuel improvement [15]. Further expected developments in describing chemical composition, remodeling of cell wall architecture and interactions between S- and G-layers should be also considered in the context of biomimetic materials [11,44].

## 5. Conclusions

Firstly, these results illustrate that Raman spectroscopy coupled with the appropriate statistical analysis of spectral data is a powerful tool for investigating cell wall structure. Secondly, they also suggest that G-layers from flax bast fibers and xylem cells, whether in flax or other species, should not be considered as identical. Differences exist both from an ontogenetical point of view, as well as from a chemical point of view. In contrast, the overall architecture and polymer distribution appear similar. From a physiological point of view, plant proprioception should be considered at the whole organism level. For flax, interpretations should not only focus on bast fibers [18], but also xylem cells (this study), as well as other cell types (e.g., parenchyma). Indeed, one could be satisfied to compare the effects of this continuous gravitropic stress on the wood by only taking the cell wall remodeling of lignified secondary walls into account. Furthermore, we should also consider that the stress signal perceived in the tension pole will modify the spectral profile of host cells. Since this signal could also be transmitted to the opposite pole, it would be interesting to compare Raman spectral profiles from stressed wood (including both opposite and tension poles) to profiles from non-stressed control wood.

## Figures and Tables

**Figure 1 biomolecules-13-00435-f001:**
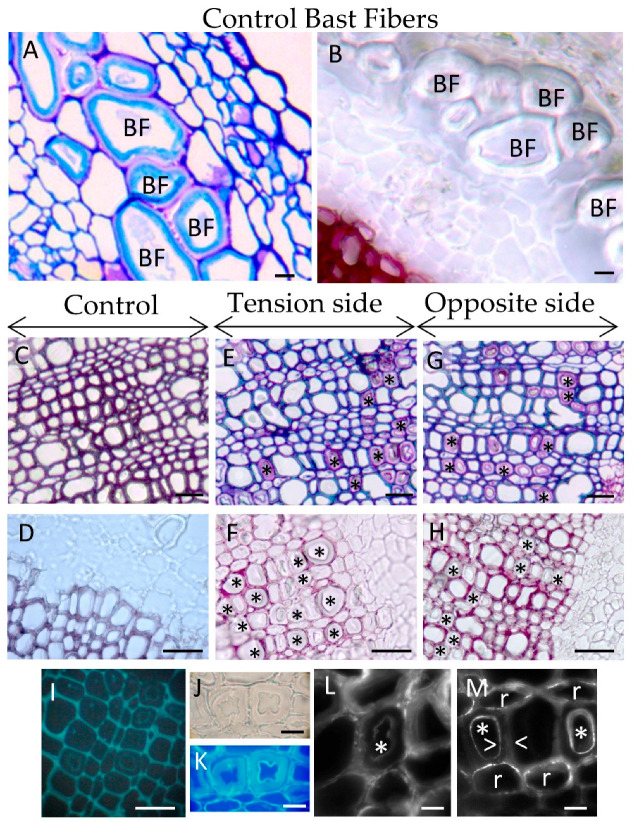
Comparison between bast fibers and xylem G-layers induced in tilted flax. (**A**,**B**) Control bast fibers (BF). No lignin is detected within BF cell wall or in cell junctions. (**C**,**D**) Control wood. (**E**,**F**) Xylem G-layer in tension side (* G-layer). It appears in grouped fiber-tracheids and rarely in ray cells. No G-layer was noticed in large vessels. (**G**,**H**) Xylem G-layer in the opposite side (* G layer). (**I**) Photonic epifluorescence only shows weak autofluorescence in the innermost part of G-layer, tension wood. (**J**,**K**) The cellulosic nature of G-layer is confirmed with calcofluor white. (**L**) Arabinan residues (labeled with LM6 antibodies) were particularly detected in the autofluorescent layer (*) (see **G**) as well as raft-shape within ray cells. (**M**) Xylan residues (labeled with LM10 antibodies) were present as dotted spots or layered-rafts both in fiber-tracheids or ray cells (r). Weak signal is detected in the autofluorescent layer of G-layer (*, white arrowheads). Semi thin section (5 μm thickness). (**A**,**C**,**E**,**G**): TBO staining, (**B**,**D**,**F**,**H**): phloroglucinol/HCl staining (Wiesner reaction). (**A**–**H**) bar = 20 μm. (**J**–**M**) bar = 5 μm.

**Figure 2 biomolecules-13-00435-f002:**
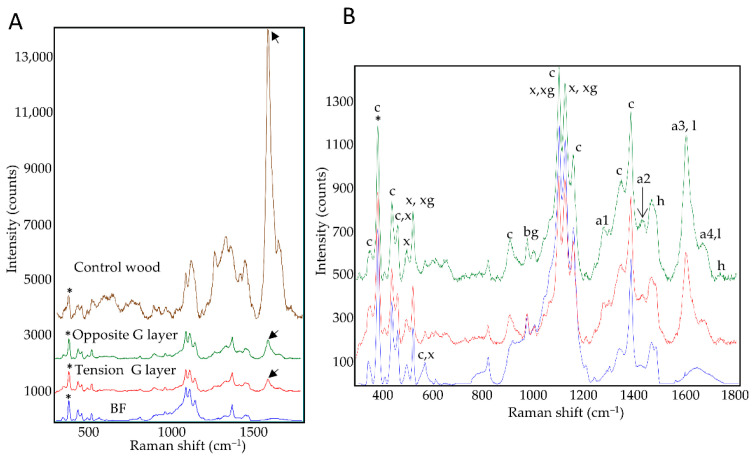
Average Raman spectra. (**A**): xylem G-layers (tension, opposite) compared to control bast fiber G-layers or control wood SCW. (**B**): zoom on average spectra from xylem and BF G-layers. tension side (red), opposite side (green), bast fibers (blue) and control wood (brown). Annotation: a1–4: aromatic compounds and monolignols; bg: beta glucan; c: cellulose; h: hemicelluloses; l: lignin; x: xylan; xg: xyloglucan. For average spectra, normalization was performed on the 380 cm^–1^ peak assigned to cellulose (*). Aromatics/lignin (black arrow). For peak assignation, see Appendix A for details [22,23,24,25,26,27,28,29]. See also Appendix A for focus on specific spectral zones (mean centered spectra).

**Figure 3 biomolecules-13-00435-f003:**
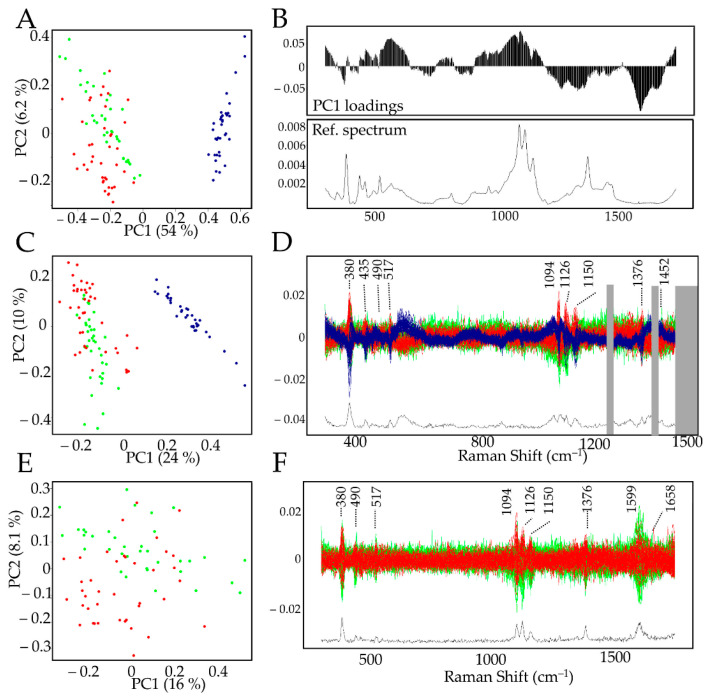
PCA analysis of xylem and BF G-layer Raman spectra. (**A**): Biplot of the full dataset. BF (blue dots) and xylem G-layer Raman spectra (red/tension and green/opposite dots) were clearly separated along the PC1 (54%) axis. Other PCs did not lead to better discrimination of xylem G-layer tension side from the opposite side. (**B**): PC1 loadings along with the average spectrum from BF. (**C**,**D**): Biplot and centered spectra of full data set modified to remove aromatics bands [1270–1421–peaks >1510 cm^–1^ ], BF (blue dots) and xylem G-layer Raman spectra (red/tension and green/opposite dots). (**D**): Xylem G-layer (red, green) and BF G-layer (blue) spectra showing removed aromatic bands (grey boxes). (**E**,**F**): Biplot and centered spectra. BF dataset was removed in order to compare only xylem G-layers (red/tension and green/opposite dots).

**Figure 4 biomolecules-13-00435-f004:**
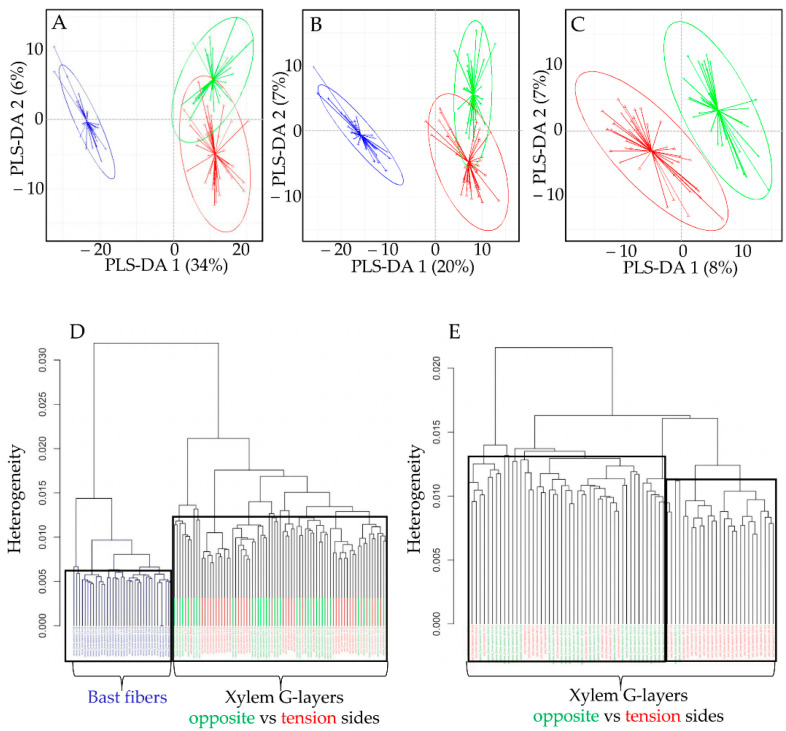
(**A**–**C**): Partial least squares-discriminant analysis (PLS-DA) biplots Raman spectra. Bast fiber (BF) dataset appears in blue, xylem G-layer datasets from opposite vs. tension sides appear in green and red, respectively. (**D**,**E**): Hierarchical clustering of Raman spectra from BF (blue) and xylem G-layers (green, red). Dendograms representing the distance (calculated as the spectral contrast angle) amongst the Raman spectra. D, black frames divided BF and xylem G-layers when the dendrogram is cut at *k* = 0.006 for BF and *k* = 0.013. E, black frames divided (incompletely) xylem G-layers when the dendrogram is cut at *k* = 0.013 mainly for opposite G-layer, and *k* = 0.012 mainly for tension G-layer.

**Figure 5 biomolecules-13-00435-f005:**
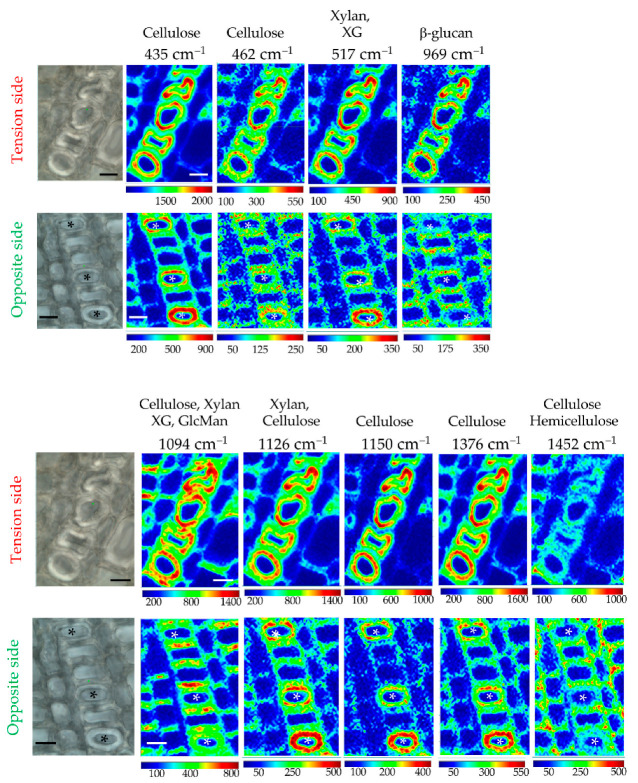
Raman chemical imaging of significant peaks. Cambial zone is located at the bottom of each image. Fiber-tracheids exhibiting a G-layer are symbolized with an asterisk. Intensities were twice as high in G-layer from the tension side than the opposite one. Compared to the opposite side, in the tension side the neighbor cells (fiber-tracheid, ray cells, large vessel) exhibit less intensity in their cell walls (particularly the periclinal wall). Bar = 10 μm. Normalization was done on the cellulose peak 380 cm^–1^. Peak assignments are given in Appendix A.

**Figure 6 biomolecules-13-00435-f006:**
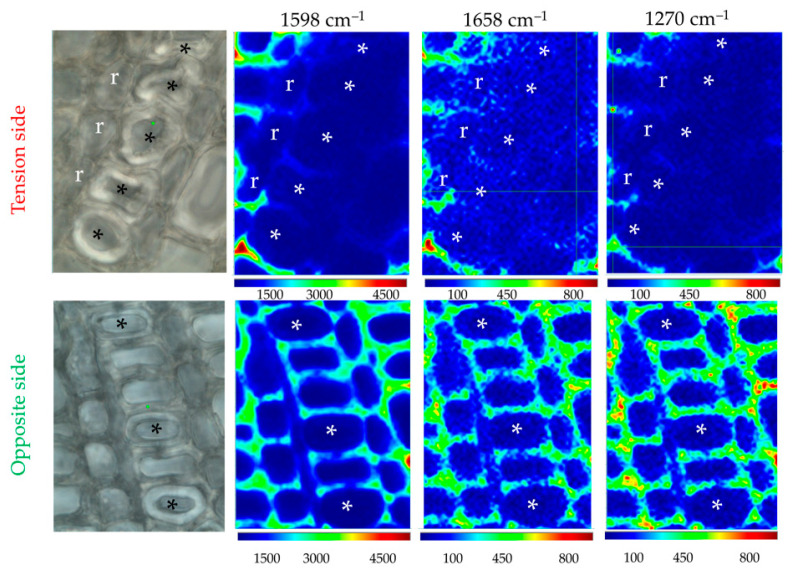
Raman chemical imaging of major aromatic/lignin peaks. This illustrates the effects of the G-layer on its closest neighbor xylem cells. The identical color scale helps the comparison of tension/opposite G-layers for a given peak. Lignin depletion is noticed in the vicinity of certain vessels (*), whatever the cell type considered (ray cells/r, fiber-tracheid or large vessel). It is particularly observed in tension side wood contrary to the opposite side. Cambial zone is at the bottom of each image. Bar = 10 μm. Normalization was done on the cellulose peak 380 cm^–1^.

**Table 1 biomolecules-13-00435-t001:** Mean confusion matrices of 100 double cross validation PLS-DA detailing classification results on the three Raman datasets. The number of spectra tested was 36, 40 and 36 for opposite, tension and bast fibers, respectively. Numbers are the absolute numbers of classifications (averaged over the 100 iterations).

	Total Datasets and All Wavenumbers: Classification Errors 3.8% *s.d*. 1.2%			
		**TRUE**		
		**Opposite**	**Tension**	**Bast Fibers**
**PREDICTED**	**Opposite**	33.39	1.71	0.00
	**Tension** **Bast Fibers**	2.61 0.00	38.29 0.00	0.00 36
	n spectra	36	40	36
	**Modified Total Datasets (Aromatics Bands were Removed):** **Classification Errors 5.8% *s.d.* 1.4%**			
		**TRUE**		
		**Opposite**	**Tension**	**Bast Fibers**
**PREDICTED**	**Opposite**	32.29	1.77	0
	**Tension**	3.71	37.20	0
	**Bast Fibers**	0.00	1.03	36
	n spectra	36	40	36
	**Partial Datasets (Only G-Layers) and All Wavenumbers:** **Classification Errors 6.6% *s.d*. 2.2%.**			
		**TRUE**		
		**Opposite**	**Tension**	
**PREDICTED**	**Opposite**	33.88	2.93	
	**Tension**	2.12	37.07	
	n spectra	36	40	

## Data Availability

All data generated or analyzed during this study are included in this published article and its supplementary information file.

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
