# Peer review of "Comparative Analysis of G-Layers in Bast Fiber and Xylem Cell Walls in Flax Using Raman Spectroscopy"

_biomolecules, 2023, doi:10.3390/biom13030435_

Round 1

Reviewer 1 Report

Dear Authors,

overall, I rate the manuscript positively. In the research work, many interesting methods were used, with the use of which the authors of the work proved their theses. I have no major comments except for minor editors, for example, table 1 should be moved to the next page, or the fact that after "Tab." or "Fig." should be a period, not a colon.

Author Response

  • Location of Table 1 : as requested in the author guidelines, the Table 1 (placed on line 277) is localized after its first mention in the text (line 270).
  • Typography of “Tab.” or ‘Fig.”: Table and Figure were written completely and numbered as mentioned in the author guidelines and also noticed in uploaded papers.
  • Table 1 was finally transformed into Table S1 (as supplemental data) as it was corrected and enriched with the references (as requested by another reviewer)

Reviewer 2 Report

The authors employed Raman spectroscopy to investigate the differences between bast fiber cell walls and xylem G-layers in flax. Although technique of principal component analysis (PCA) was introduced, the scientific novelty of this work is very limited and not enough for publication. The differences between the gel layer and the normal secondary wall were found early. This paper does not give more information. It is known that the biggest differences between the Raman spectra of the G-layer and that of secondary wall cells is the characteristic peak of lignin around 1600 cm-1. They can be easily distinguished by mapping 1600 cm-1. I don't think it is necessary to use PCA to separate G-layer from secondary wall. The difficulty is to identify the other cell wall layers, such as compound middle lamella, cell corner and secondary wall. However, this work has been done by previous study. By the way, G-layer is not a cell type of plant cells. It is just a special part of cell wall. Therefore, I suggest to reject paper in the present form.

Author Response

The reviewer is completely correct if we consider the typical secondary walls of lignified cell walls (e.g. xylem vessels, tracheids, fibers etc.). However, in flax bast fibers certain authors (see publications by the group of Tatyana Gorshkova) consider that an additional layer (G-layer) is produced interior to the S2 layer of the secondary cell wall (i.e., the flax fiber secondary cell wall would be composed of S1, S2 and G layers, in contrast to the S1 and S2 layers of lignified cells. Subsequently, the same authors have argued that the flax fiber G layer is equivalent to the G-layer found in tension wood xylem cells. In our paper we compared the G-layers from flax bast fibers with G-layers from flax xylem cells, we did not compare flax bast fiber G-layers with the lignified secondary cell wall. Furthermore, the group of Tatyana Gorshkova have also suggested that a new class of “tertiary cell walls” should be defined by grouping cellulosic bast fibers (e.g. of flax) and xylem G-layers (a stress-induced, cellulosic layer produced after formation of the secondary lignified cell wall).

The scientific question addressed in this paper was therefore to see whether this hypothesis could be validated or not using Raman spectroscopy. The use of this technique provided a much more detailed characterization of the composition of flax bast fiber and xylem G-layers in stressed plants in comparison with a previous study (See Petrova et al., 2021, ref.12 in MS) that was based on the use of only 2 antibodies. As our results show, these two cell wall layers are similar but NOT equivalent. Furthermore, our paper also provides novel information about wall remodeling in the neighboring cells adjacent to xylem cells containing G-layers.  We therefore believe that our work does bring new information about stress-induced changes to cell wall structure in flax, as well as illustrating how Raman spectroscopy and chemical imaging, associated with appropriate statistical treatment of data could be used to explore this phenomenon in other fiber plant species.

The reviewer is also correct in stating that the G-layer is not a cell type, we have carefully verified and modified the text where necessary to avoid any possible confusion on this point

Reviewer 3 Report

1.     The authors should provide justification for their decision to use only cell walls of Bast fibers to compare with xylem G-layers but not secondary walls of “control wood (A,B)” from Figure 1; L87-89 are not successful clarifying the selection of tissues for analysis

2.     Fig. 1C-E in reference to the text in L229-235 should highlight the cells with G-layer; for example, asterisks might be used to mark cells with a G-layer as in Figure 5

3.     Figure 1 should show preliminary characterization of BF with some of the tests used for preliminary characterization of xylem G-layers – TBO and Wiesner, for example

4.     L 147 onwards, “Average spectra were obtained for three different fiber/G-layer categories (BF secondary cell wall, tension wood and opposite wood G-layers) and normalized on the 380 cm-1 peak assigned to cellulose.” The authors failed to describe how the average spectra were obtained; are these the average from three measurements recorded at three different locations in individual studied tissues or these are the average of three measurements made at one location in these tissues or these are the average of three measurements recorded for three individual plants/control and stressed. In addition, Figure 2 should include Raman spectra of control wood used in characterization of samples A and B in Figure 1 for comparison

5.     Fig. 2: the spectra seem to have been normalized on a peak that has an asterisk (*) which is not the 380 cm-1 peak. The latter peak is the sharp peak next to it. The peak with the “*” seems to be 352 cm-1 feature which is quite weak and cannot be used for normalization. This is supported by the fact that, as shown, the three 380 cm-1 peaks have different intensities (Fig. 2).

6.     Several of the band assignments in Table 1 are to be reviewed/checked. In the fingerprint region, xylan has a band at 494 cm-1 that does not overlap with other Raman bands (refer to assignments reported in for example Agarwal, U.P., Ralph, S.A. (1997): “FT-Raman spectroscopy of wood: Identifying contributions of lignin and carbohydrate polymers ibn the spectrum of black spruce (Picea mariana),” Applied Spectro 51 (11) 1648-1655). Hence, when it is not detected in a spectrum, it is highly unlikely that other xylan bands will be seen.  490 cm-1 contribution is a xylan band not one of cellulose, as assigned in Table 1. Similarly, 517 cm-1 is a cellulose and xyloglucan band; not due to xylan, 562, 607, and 650 cm-1 band assignments are also to be checked (cite the references on which these are based), 1126 cm-1 is due to both xylan and cellulose not just the former; 1203 cm-1 is not a peak of cellulose;  1452 cm-1 is due to both cellulose (amorphous) and hemis; and lastly, 1599 and 1658 cm-1 should have been assigned to lignin and not to the listed lignin models.

7.     In light of the comments above, multivariate analysis has to be reviewed. The spectral data set needs to be straightened out first.

8.     Fig. 5: see comments about band assignments in Table 1

9.      Figure 5: how were Raman chemical images produced- based on absolute intensity or baseline corrected intensity? Also, comparing Fig.5 and Fig. 6 plots, why the lignin-based images have much higher counts (1500 to 4500 compared to 200 to 550 in Fig. 5)? Especially, considering that the lignin region bands are not all that stronger compared to the cellulose peaks.

10.  In Fig. 5A and B, where G-layers are symbolized with asterisks, in most of the images, there is a lower intensity layer (in green) on the lumen side compared to the layer further away from lumen (red and yellow in color) (e.g., plot of 1094 cm-1 band)? Why such differences in intensity distribution and what does that imply?

11.  Although, in Table 1, bands are assigned to xyloglucan, most of its bands overlap heavily with that of cellulose. Therefore, how does one know it is present in flax? Is there a chemical analysis available? If yes, that should have been referred to.

12.  The results of chemical composition analysis of bast fibers and xylem fibers from control plants and stressed plants would be beneficial to support the conclusions of this work.

13.  Ref. #3 = Ref. #11 Gorshkova et al. 2018 doi: 10.1111/nph.14997

14.  Origin of rabbit primary antibodies L124-125 and goat anti-rat antibody L130

15.  Line 163 “variablility”

Author Response

  • The Introduction was reformulated in order to more clearly explain why we focused our attention on comparing G-layers from flax bast fibers G-layers in flax xylem.
  • Asterisks have been added on Figure 1, C-E.
  • Bast Fibers histochemistry through TBO and Wiesner reaction was added to the Figure 1.
  • Average spectra: the method was reformulated in the dedicated §.
  • Figure 2: it was an error during figure annotation. Asterisk has been moved to the next peak (major one) at 380 cm-1.
  • Table 1 and checking assignment: Table 1 was changed into Table S1 as new informations such as references for each peak, was added. Dual assignment (for example, Cellulose/Xyl) was already indicated on Figure 2. But it is now also mentioned in Table S1.
  • Multivariate analyses were checked according to the previous corrections (notably Table S1)
  • Figure 5 was corrected according to Table S1
  • Figure 6, lignin intensities: Raman imaging was performed according to baseline corrected intensity. We have checked the lignin map, and this intensity scale was due to lignin high accumulation in cell corner and other periclinal/anticlinal cell walls (see up left edge of the picture). So, in the part of the section showing G-layer, the xylem cell wall was effectively relatively deprived in this chemical bond.

For the other chemical imaging, notably for cellulose, the cells present in the map area exhibited the same range of intensity. Even if, such intensity scale was different according to the pole (tension, opposite).

  • Figure 5A, 5B, intensity distribution: when we apply a vector in order to acquire spectra every micron, from the middle lamella (from xylem cell wall) to the cell lumen, crossing through G-layer, we have a curve with a maximum corresponding to the orange/red pixels. Effectively, close to the lumen, the ratio is lower. 

    Figure 5A, 5B, interpretation: such progressive and symmetrical abundance (as concentric layers) of some chemical bonds, is probably due to the ontogeny of this G-layer. It appeared, probably in a non-autonomous manner in fully differentiated cells (i.e. by diffusing precursors from the middle lamella, through primary, then the different secondary cell wall layers). Its deposition as a post mortem event, is compatible with the later deposition of extractives such as terpenoids, alkaloids or phenolic compounds that could explain the autofluorescence of the innermost part of the G-layer (Figure 1I, Figure 2B a1-a4 peaks).

  • in our system, the xylem G-layer host cells (fiber-tracheids) were in the process of completing a cell death program when the stress conditions (tilting) were applied. The G-layer is therefore produced during the secondary cell wall patterning, and/or will replace/alternate with the S3 layer (Clair et al., 2018; Ruelle et al., 2007; Ghislain et al., 2016, Refs 16-26-27 in this MS). We can also suggest that such progressive intensity and symmetrical depositions could help the stressed plant to mechanically recover to a vertical position according to its gelatinous status. Clair et al., 2018 suggested that such a G-layer exhibited high mesoporosity and ability to generate large contraction of the layer along the axis fiber. If the microfibril angle is abruptly changed, such a G-layer could detach easily from S2-layer (i.e. from fiber-tracheid) and induce a release of the high tensile stress.

    • Presence of Xyloglucan in flax as it overlaps with cellulose: The presence of xyloglucan in flax fibers was demonstrated in previous papers (Chabi et al., 2017; Rihouey et al., 2017, Refs 6-2 in this MS)

    • The reviewer is correct - in theory. While it is true that chemical analysis of xylem G-layers and bast fiber G-layers would provide valuable support to the conclusions reached in this paper, in practice this would be very difficult. “Wet chemistry” techniques require a minimum amount of material that is ground-up for subsequent polymer/sugar extraction. Although the different cell types (bast fibers, xylem cells) could be obtained by (micro)-dissection, chemical analysis on ground-up cell wall material would only provide information on global cell wall composition (i.e. it would be an average of all cell wall layers - middle lamella, primary cell wall, secondary cell wall, G-layer). To get around this problem, we would therefore have to obtain sufficient quantities of G-layers from xylem cells and from bast fibers. Although it has been reported that xylem G-layers can be separated from xylem vessels through sonication (Nishikubo et al., 2007, Ref 32 in this MS), no such method currently exists for bast fibers where the G-layer is more tightly associated to the secondary cell wall. Furthermore, since chemical analysis is performed on ground-up material, spatial information on polymer distribution within the different cell wall layers is lost. This is why we decided to implement an in-situ imaging technique. 

    • It was also curious for us. I have click on the false Gorshkova’s reference (2018 instead of 2022) with Mendeley bibliography software. It is now corrected and new references were added according to the referee suggestions.
    • The company that provides primary and secondary antibodies, and the host organisms, were already mentioned. We have added new information about the antigen used to induce antibody production, as well as the epitope targeted.
    • Line 163, error in the term “variability” was corrected.
  •  

Reviewer 4 Report

This paper is well-structured, informative, and novel. However, it requires some modifications to become acceptable for publication. 

1. The knowledge gap is not clearly presented in the introduction. The authors should describe the knowledge gap and explain how this study bridges this gap.

2. I suggest the authors add more information regarding the difference between normal (opposite) wood and reaction (compression/tension) wood. The following paper may help.

Tarmian, A., Sepeher, A., & Rahimi, S. (2009). Drying stress and strain in tension wood: A conventional kiln schedule to efficiently dry mixed tension/normal wood boards in poplar. Drying Technology27(10), 1033-1040.

3. Research limitation is not presented. It must be mentioned in the discussion and conclusions.

4.  Deliverables (practical implications) of this study are not discussed. I strongly suggest adding implications to the conclusions.

5. Recommendations for future studies should be mentioned in the discussion and conclusions.

Author Response

  • Knowledge gap: new informations were added in the Introduction §.
  • Add information about tension/opposite= normal, woods. This study aims to explore the hypothesis of whether flax BF and xylem G-layers should be integrated in the “tertiary cell wall” class as suggested by some authors, due to their gelatinous status. Such cell wall layers are highly cellulosic and poor in lignin compared to the primary and secondary walls of both normal, opposite and tension wood cells. As such, the information provided in the introduction and discussion of this paper does not go into detail about the structure of these wood types. Nevertheless, as requested by this referee, information about these two types of wood has been added.
  • Research limitations: Advantages and technological or statistical limits have been indicated withing the text (end of the § Discussion).
  • Deliverables and practical implications:

New informations about practical implications are added in the Discussion.

  • Future studies, perspectives: see the second point. Perspectives have been added to the conclusion.

Round 2

Reviewer 3 Report

accept the revised version.

Reviewer 4 Report

The manuscript has been revised satisfactorily. So, the paper in its current format can be accepted for publication.